# Polymeric Nanoparticles-Loaded Hydrogels for Biomedical Applications: A Systematic Review on In Vivo Findings

**DOI:** 10.3390/polym14051010

**Published:** 2022-03-02

**Authors:** Débora Nunes, Stéphanie Andrade, Maria João Ramalho, Joana A. Loureiro, Maria Carmo Pereira

**Affiliations:** 1LEPABE—Laboratory for Process Engineering, Environment, Biotechnology and Energy, Faculty of Engineering, University of Porto, Rua Dr. Roberto Frias, 4200-465 Porto, Portugal; deborasspinola@hotmail.com (D.N.); stephanie@fe.up.pt (S.A.); mjramalho@fe.up.pt (M.J.R.); 2ALiCE—Associate Laboratory in Chemical Engineering, Faculty of Engineering, University of Porto, Rua Dr. Roberto Frias, 4200-465 Porto, Portugal

**Keywords:** nanomaterials, polymers, drug release, local delivery, administration routes, thermosensitive hydrogel, cancer therapy, chronic wound treatment

## Abstract

Clinically available medications face several hurdles that limit their therapeutic activity, including restricted access to the target tissues due to biological barriers, low bioavailability, and poor pharmacokinetic properties. Drug delivery systems (DDS), such as nanoparticles (NPs) and hydrogels, have been widely employed to address these issues. Furthermore, the DDS improves drugs’ therapeutic efficacy while reducing undesired side effects caused by the unspecific distribution over the different tissues. The integration of NPs into hydrogels has emerged to improve their performance when compared with each DDS individually. The combination of both DDS enhances the ability to deliver drugs in a localized and targeted manner, paired with a controlled and sustained drug release, resulting in increased drug therapeutic effectiveness. With the incorporation of the NPs into hydrogels, it is possible to apply the DDS locally and then provide a sustained release of the NPs in the site of action, allowing the drug uptake in the required location. Additionally, most of the materials used to produce the hydrogels and NPs present low toxicity. This article provides a systematic review of the polymeric NPs-loaded hydrogels developed for various biomedical applications, focusing on studies that present in vivo data.

## 1. Introduction

The physiological barriers of the human body challenge the traditional delivery of drugs, limiting drug access to the desired organs and tissues. Furthermore, the efficacy and retention of drugs in the target tissue are affected by their bioavailability, pharmacokinetic, and pharmacodynamic parameters. These challenges lead to the need for higher dosages and more frequent administrations to reach effective treatment doses, which induces undesirable side effects and toxicity in the other tissues [1].

Over the last years, due to considerable advances in the nanotechnology field, several nanomaterials have been developed as drug delivery systems (DDS). These systems have emerged to overcome the drawbacks of drug administration by improving the drug’s solubility and bioavailability, decreasing drug degradation, and extending the drug’s half-life [2,3]. DDS can provide a specific and targeted therapy, reducing the required dose to achieve a therapeutic effect. DDS capabilities make therapies more accurate, effective, and less invasive by preventing systemic toxicity and unwanted side effects [2,3,4,5]. Among the potential DDS approaches under development, the design of nanoparticles (NPs), hydrogels, and, more recently, NPs-loaded hydrogel (NLH) systems for drug release applications are gaining attention.

The growing advances in the use of nanomaterials led to the need to establish guidelines and regulations for their usage in medicine. Updates in the future are expected to guarantee the quality, effectiveness, and security of these DDS for human use [6]. Besides, scale-up manufacturing is another important condition for clinical use and commercialization of DDS. The clinical application of the DDS faces some concerns regarding their production. When it comes to large-scale production, the procedures should ensure the preservation of the DDS physicochemical characteristics for the desired application since these properties directly affect the efficiency, safety, and drug delivery capabilities of the developed DDS. Simultaneously, the methods must be scalable, reproducible, and cost-effective [7]. Good manufacturing practices verification is required to ensure conformance in large-scale production [6]. In addition, to evaluate whether DDS large-scale production influences clinical performance, comprehensive quality controls of drug carriers are essential [8]. Finally, clinical trials are mandatory to determine the ratio between benefits and risks of the developed DDS [6].

As in vitro studies can only give a partial indication of potential toxicity, in vivo studies are fundamental to evaluate the efficacy and safety of these DDS [4]. In this sense, this review aims to discuss the most recently developed polymeric NLH systems for biomedical applications focusing on their in vivo performance for different administration routes, including parenteral and topical administration. Polymeric NLH systems integrate NPs into a hydrogel, synergistically combining their individual functionalities and benefits. The resulting system presents increased performance and can be successfully employed as DDS for biomedical applications, including tissue engineering, bacterial applications (e.g., wounds or eye infections), and contact lenses [9,10].

Several review papers focusing on the use of NPs combined with hydrogels have been published in recent years owing to the importance of this subject for the scientific community [11,12,13,14,15,16,17,18]. Despite the considerable content provided, while some of these works only focus on NLH production methods, others focus on applying NLH systems to treat specific disorders. Thus, the current work provides the first systematic review of this area, comprising all the research evidence published in the period between 2011 and 2021. The research methodology of this systematic review is described in Section 2, which presents the protocol used to select the articles. Section 3 introduces polymeric NLH systems, and Section 4 describes the routes of drugs administration, divided into sub-sections, with a summary of the outcomes of the selected works for each route. Finally, the discussion and conclusion are presented in Section 5, summarizing the developed work and the perspective based on the obtained findings.

## 2. Research Methodology

This systematic review is based on the PRISMA (Preferred Reporting Items for Systematic Reviews and Meta-Analyses) reporting guide [19], a process that identifies, evaluates, and interprets all published research relevant to the study topic. An initial literature search of polymeric NPs combined with hydrogels was performed. Databases PubMed, Science Direct, Google Scholar, Scopus, and Web of Science were consulted between November 2021 and December 2021, considering “hydrogel”, “polymeric nanoparticles”, “nanoparticles”, “nanogels”, and “gel” as key terms for all databases. Articles were then collected based on their main themes and relation to the current article’s scope.

The inclusion criteria applied was polymeric NLH developed for biomedical applications, published in the period between 2011 and 2021. Only articles in English were included. As exclusion criteria, studies that were repeated in different databases, studies published in journals that were not Quartile (Q) 1 or Q2, review articles, and research that did not evaluate the in vivo performance of the developed polymeric NLH. As the journal’s quartile was an exclusion criterion, all the considered studies were published in indexed journals of renown, indicating good paper quality.

Based on the applied research strategy, 119 articles were found, including 33 articles in PubMed, 20 articles in ScienceDirect, 37 articles in Google Scholar, 19 articles in Scopus, and 10 articles in Web of Science. After duplication analysis, 54 studies were eliminated, leaving 65 articles. Inclusion and exclusion criteria were applied, removing 28 articles that did not meet the research requirements. After full-text article evaluation, 7 were excluded by lack of information or corresponding to a hydrogel formed by the NPs themselves, rather than NLH, leaving 30 articles for the qualitative synthesis. This methodology is summarized in Figure 1.

The expected outcome was identifying an improved activity of NPs and hydrogels combined as DDS. In in vivo experimental studies, the description of the results was mainly based on the performance of polymeric NLH as DDS to explore new therapeutic strategies. The following data were extracted from the included studies: NPs material, hydrogel material, NPs and hydrogel production methods, description of the drug-loaded into polymeric NLH systems, biomedical application, route of administration, in vitro and in vivo studies and results; and are then discussed under further sub-sections.

## 3. NPs-Loaded Hydrogel System

NPs are colloidal structures designed and produced to transport drugs across biological barriers. Their optimal size range is approximately between 100 and 200 nm. They protect drugs from degradation, increasing their half-life, improving drugs’ bioavailability, and providing a sustained and localized release [13]. Moreover, their surface could be modified for targeted delivery, reducing drugs’ toxicity resultant from the systemic spread and, consequently, side effects [20]. Among them, polymeric NPs have been widely chosen as DDS in several biomedical applications due to their high stability, water solubility, biocompatibility, biodegradability, and non-immunogenicity [21,22]. Another advantage of these NPs is their capacity to encapsulate both hydrophilic and hydrophobic drugs [20]. Moreover, their loading capacity, drug release kinetics, and biological performance could be regulated by adjusting their composition or surface charge. However, these NPs are quite predisposed to premature burst release of drugs [13,22], their permanency on the target site until complete drug release is unpredictable [13,23]. When in contact with the biological environment, NPs can present instability or clearance by the immune system [5].

Hydrogels are three-dimensional porous structures produced with hydrophilic polymers through physical or chemical cross-linking methods [24] and can be prepared from a wide range of natural and synthetic polymers. Natural polymers include alginate, chitosan, gelatin, collagen, hydroxypropyl methylcellulose (HPMC), and hyaluronic acid (HA); in contrast, synthetic polymers could be polyacrylamide (PAM), poly(hydroxyethyl methacrylate) (PHEMA), polyvinylpyrrolidone (PVP), poly(vinyl alcohol) (PVA), poly(ethylene glycol) (PEG), and poly-ε-caprolactone (PCL) [25]. Hydrogels are materials with great solute permeability, and a high-water retention capability [9,25,26,27]. Depending on the polymers employed in their production, hydrogels can be biocompatible, biodegradable, and present minimal toxicity. They can encapsulate molecules in an effective amount, protecting and releasing them over time while increasing their local concentration and reducing their toxicity in the remaining tissues [28]. Moreover, hydrogels kinetics can respond to biological, chemical, or physical external stimuli. Biological triggers include antibodies or enzymes; chemical factors comprising pH, type of ions, or organic solvents; and physical stimuli include temperature, light, and magnetic or electrical fields [25]. Thermo-sensitive hydrogels are the most commonly used variety of hydrogels for medical use due to their sol-gel transition behavior at body temperature. While at room temperature hydrogels are in the form of an aqueous suspension that can be easily injected; at body temperature, the solution rapidly transits to a stable gel [25,29]. Their administration is minimally invasive since most of them can be administered without the need for a surgical procedure [11]. Besides, thermo-sensitive hydrogels manage to mold themselves perfectly to the shape of the place where they are administered, creating a drug depot for a localized and sustained release [1]. On the other hand, hydrogels have a few drawbacks limiting their application as DDS. Hydrogels have weak mechanical properties; for example, their mechanical strength decreases after swelling [19]. Drug release from hydrogels depends on their network structure, rearrangement, size, materials employed in their production, and drugs’ physicochemical properties [26]. Most of the time, hydrogels present an initial burst release of drugs when in contact with the release medium due to their high-water content [15,30]. Another problem of hydrogels’ hydrophilic nature is the integration of poorly soluble drugs, which are rapidly released through diffusion [13,31].

Individual limitations of NPs and hydrogels could be addressed with their combination into a single platform (Figure 2). As NPs can be physically or covalently integrated into hydrogels, the development of these DDS has emerged, taking advantage of their benefits synergistically combined in one system [26].

Besides the biocompatibility, biodegradability, and non-toxicity of both NPs and hydrogels as individual DDS, their combination provides benefits that none of them could achieve independently (Table 1).

NPs and hydrogels can both provide multiple drug loading. When combined, the hydrogel protects the NPs from degradation, prevents their aggregation [33], and promotes the local delivery of drugs [15,30]. Incorporating NPs into hydrogels can improve hydrogel mechanical properties, such as strength, stiffness, and degradation, in a concentration-dependent relationship [13]. NPs may also act as a hydrogel crosslinking and fortify the stimuli-responsive behavior of the hydrogel [12]. Although NPs and hydrogels can improve drugs’ bioavailability and release them over time [13], their combination forms a depot at the administration site for prolonged local drug retention [32]. This dual delivery system provides a double encapsulation of drugs, regulates drug release kinetics, and prevents the initial burst release. It can also encapsulate both hydrophobic and hydrophilic drugs [13,17]. These benefits work together to minimize side effects and systemic toxicity [13] and improve the therapeutic effect of treatment and patients’ compliance [15,30].

## 4. Routes of Drugs Administration

The administration route determines the effectiveness of the treatment and depends on the drug characteristics, biomedical application, and patients’ compliance. The routes of drugs administration are classified as enteral, parenteral, topical, or inhalation (Figure 3) based on the drug’s administration site.

In the enteral route, drugs are absorbed by the gastrointestinal tract and comprise oral, sublingual/buccal, and rectal administration. Oral administration of drugs is the most frequently used administration route in ambulatory [34]. This route is convenient for frequent treatments that require repeated intakes. Besides the easy application, safety, convenience, and economic advantages of this administration, some limitations compromise the therapeutic effect. Due to first-pass metabolism and poor drug pharmacokinetics and pharmacodynamic properties, such as low oral absorption, low bioavailability, and enzymatic degradation [34], the drug concentration is highly reduced until they reach their action site. Another limitation of this route of administration is the poor targeting of drugs [35], which slows their onset of action and induces systemic toxicity.

For the parenteral route, drugs directly reach the systemic circulation and are usually administered with an injection. This route is preferable for drugs with poor solubility and stability in the gastrointestinal tract. The commonly injections forms are (a) intravenous, where the drugs are directly administered to the systemic circulation for a rapid drug effect; (b) intramuscular, where the drugs are directly administered to the muscle, that due to its high blood circulation, provides a higher absorption of drugs; and (c) subcutaneous, where the drugs are administered in the subcutaneous tissue that, due to its reduced vascularization compared to intramuscular injection, provides a slower absorption of drugs, suitable for treatments that require frequent administrations. However, drugs can also be administered through other less common injection forms, such as intradermal, intraperitoneal, intrathecal, or local injections [34]. The limitations of these routes include systemic toxicity and subsequent side effects depending on the drugs [34].

The topical route comprises the administration of drugs at the skin and mucous membranes, such as eyes (through instillation or irrigation) or vagina. The topical application through the skin is not only painless and non-invasive but is also simple to apply, resulting in high patient compliance. It also allows increased amounts of drugs at the target site. The topical route avoids the first-pass metabolism, which improves drugs bioavailability and treatments efficacy [36]. The drawbacks of this route depend on each mucosa. The skin has an acidic pH (4.2–5.6), requires a high degree of hydration, and presents a barrier at the most outermost layer that could limit drugs’ permeability [36]. Due to the aqueous environment of eyes and their tear drainage, drug loss is frequent, which induces low retention of drugs at the ocular tissue. Furthermore, the drug’s low bioavailability contributes to an ineffective treatment [34]. Vaginal administration efficacy is reduced by degradative enzymes and vaginal fluid, decreasing the drugs’ bioavailability and residence time [36].

The inhalation route comprises the nasal administration of drugs through vaporization, gas inhalation, or nebulization and could be applied for both local and systemic therapies. The drugs are rapidly absorbed into the respiratory epithelium due to their large surface area and huge permeability [37]. Then, due to respiratory epithelium’s high irrigation, drugs could be directly diffused to the bloodstream, avoiding the first-pass metabolism and enzymatic degradation, which increases their bioavailability [34]. Besides, this route provides a target delivery in the case of pulmonary problems. Compared with other administration routes, inhalation provides fewer systemic side effects. However, the drug’s efficacy can be conditioned by the size of the inhaled molecules, physiology of the patient’s respiratory system, mucosal turnover, and nasal epithelium that reduces the nasal residence time of drugs [34,36,37].

Given the limitations of current medicines, NLH systems can be advantageous for therapies that require a local, frequent, and long-term administration of drugs. The most appropriate route of administration is chosen based on the biomedical application. The following sections provide and discuss the most recently developed polymeric NLH for the various administration routes, including subcutaneous, local, ocular, and topical.

### 4.1. Parenteral Administration of Polymeric NPs-Loaded Hydrogels

#### 4.1.1. Subcutaneous Administration

Drug solutions are quickly dispersed through neighboring tissues in subcutaneous administration and consequently eliminated. Most of the time, regular administrations are required to maintain the drug levels in the blood and achieve the desired therapeutic effect. Furthermore, subcutaneous administration is associated with high dosages of drugs to reach therapeutic levels, leading to undesired side effects [38]. Considering this, polymeric NLH systems have been investigated for therapies involving drugs’ subcutaneous administration (Table 2). These systems ensure a controlled and sustained drug release, significantly decreasing their frequency of administration. Besides, NLH systems can maintain the therapeutic levels of drugs, thus increasing their efficacy [39].

Peng et al., (2013) proposed poly(3-hydroxybutyrate-co-3-hydroxyhexanoate) (PHBHHx) NPs incorporated in a thermosensitive hydrogel for an ultralong sustained release of insulin for diabetes management [40]. The hydrogel was prepared with chitosan and β-glycerophosphate disodium salt (β-GP) as a crosslinking agent, and their concentrations were optimized to obtain gelation at 37 °C. In vitro studies proved that the incorporation of the NPs did not affect the hydrogel’s gelation temperature, gelation time, or degradation. The performed in vitro release studies revealed the significant role of the NPs-hydrogel combination in prolonging insulin release, compared with free insulin or insulin-loaded hydrogel. Further, to evaluate the in vivo performance, animal experiments were performed with diabetic transgenic mice. The animals were divided into three groups and subcutaneously injected with control hydrogel, insulin-loaded hydrogel (4 IU/kg), and insulin-loaded NLH (6 IU/kg) separately. Compared to the other groups, the animal group administrated with the insulin-loaded NLH presented a prolonged hypoglycemic effect, supporting in vitro release studies that showed that the combination of NPs with hydrogel exhibited a more sustained insulin release. The authors concluded that the NLH prolonged the insulin release and the hypoglycemic effect for more than 5 days after a single subcutaneous administration, which allows for a reduction in the frequency of insulin administration.

Subcutaneous applications of these NLH systems have also been widely explored for other biomedical applications by increasing drugs’ bioavailability and improving therapies, such as in the case of cancer treatments, as proposed by Sun et al. (2021) [28]. The authors developed an injectable thermosensitive hydrogel containing poly(lactic-glycolic acid) (PLGA) NPs as a synergic approach to destroy the tumor’s extracellular matrix. The hydrogel was composed of PCL–PEG–PCL, and PLGA NPs were used to encapsulate booth photosensitizer indocyanine green (ICG) and nitric oxide donor l-arginine (l-Arg). The ICG is a photosensitizer able to produce reactive oxygen species under infrared light irradiation, which will cause apoptotic cell death and, simultaneously, the oxidation of l-Arg, which will generate nitric oxide and inhibit cancer cell proliferation. The in vitro studies showed that the incorporation of the NPs into the hydrogel did not change the sol-gel transition behavior of the hydrogel. The authors also verified that loading the NPs in the hydrogel increased the ICG and l-Arg concentration and retention at the tumor site, decreasing their toxicity in the other tissues. In vitro experiments were performed with human mammary carcinoma cells to evaluate the cellular uptake, cytotoxicity activity, and apoptosis effect of NLH, compared with NPs in solution and ICG in solution. Cellular uptake assay revealed efficient endocytosis of NPs by cancer cells. In vitro apoptosis showed higher apoptosis of cancer cells for the NLH formulation than NPs in solution and ICG in solution. The in vivo studies with mammary carcinoma-bearing transgenic mice showed tumor growth inhibition and regression of the established tumors when the animals were subcutaneously administered with the NLH.

For multiple myeloma therapy, Lee et al. (2018) proposed the administration of bortezomib (BTZ) through its encapsulation into NPs, and the incorporation of those NPs in a hydrogel [41]. The NPs were produced with a triblock copolymer of PEG-phenylboronic acid-polycarbonate. The hydrogel was prepared with P(Bor)5-PEG-P(Bor)5 and P(Gu)5-PEG-P(Gu)5 polymers and optimized to obtain suitable properties for injection and BTZ-loaded NPs delivery. The incorporation of NPs in the hydrogel increased the storage moduli but did not affect the injectability. In vitro release was investigated at pH 7.4 and 5.8, to mimic the extracellular and endolysosomal environments, respectively. Results showed a faster BTZ release in the acid environment than pH 7.4. A multiple myeloma xenograft mouse model was used to evaluate the in vivo anti-cancer activity of the hydrogel. Animals were injected subcutaneously at approximately 1 cm away from the tumor site (0.8 mg/kg BTZ, 150 μL). Results showed a hydrogel degradability of more than 14 days and an excellent tolerance to the formulation. The anti-cancer efficacy was investigated by comparing animals injected with BTZ-loaded NPs incorporated in hydrogel and BTZ-loaded NPs in solution. Mice treated with BTZ-loaded NLH formulation showed smaller tumor size and inhibition of tumor progression.

#### 4.1.2. Local Administration

Local administration was envisaged to allow for the accumulation of drugs at the target tissue, enhancing treatment efficacy while minimizing systemic toxicity. Thus, several polymeric NLH have been proposed for local drug delivery (Table 3) since they can reduce drugs’ clearance and increase their retention time [42]. By achieving higher drug concentrations at the target tissue, hydrogels can prolong the therapeutic activity of drugs and reduce their systemic toxicity. Thermosensitive hydrogels attract particular attention for local implantation due to their ability to increase viscosity above physiological temperatures, allowing for their injection in the liquid form and further gelation in situ [38].

Thus, thermosensitive hydrogels have been widely explored for intratumoral drug delivery. Chemotherapy is associated with several serious side effects, and local delivery by hydrogels can overcome those. For instance, Thakur et al. (2020) developed an in situ injectable NLH for carboplatin delivery, a drug used to treat various cancers, such as breast, cervix, colon, lung, prostate, among others [43]. Carboplatin was entrapped in ethyl cellulose NPs to overcome the toxicity issues of the drug. Then, the loaded NPs were incorporated in a thermosensitive hydrogel composed of chitosan. The experimental parameters were optimized to obtain the ideal gelation time, temperature, and syringeability for the NLH. The authors also verified an improvement in carboplatin in vitro release profile, with a sustained and controlled release for drug-loaded NLH, compared with drug-loaded hydrogel, drug-loaded NPs in solution, or pure drug. In vitro cytotoxicity was performed with the drug-loaded NLH and carboplatin commercial formulations against six different human cancer cells: cervix, colon, prostate, lung, osteosarcoma, and normal breast epithelial cells. For all cell lines, no cytotoxicity was observed. Furthermore, in vitro cellular uptake study was performed in human osteosarcoma cells, whose results proved that the NPs could penetrate those cells. Further, studies were performed with carcinoma mice models to evaluate the in vivo efficacy of the developed NLH. The animals were injected intraperitoneally with the drug-loaded NLH at 123 mg/kg. The results revealed a significant improvement in the drug’s pharmacokinetic profile, which leads to a reduction of systemic toxicity. The authors concluded that the drug-loaded NLH provides a great anti-tumor activity with a decrease in tumor size, justified by the higher NPs’ uptake by cancer cells and the NLH’s ability to release the drug in a sustained and controlled manner for 7 days.

An approach for hepatocellular carcinoma treatment was proposed by Gao et al. (2021) [44]. The authors developed an injectable thermosensitive hydrogel composed of Pluronic F-127 for intratumoral administration. The hydrogel was co-loaded with norcantharidin-loaded NPs and doxorubicin, two widely used anti-cancer drugs that potentiate the treatment efficacy when administrated together. The NPs were prepared with poly(ε-caprolactone)-poly(ethylene glycol)-poly(ε-caprolactone) (PCL-PEG-PLC) and used to encapsulate norcantharidin due to its poor water solubility and low bioavailability. The in vitro release studies proved that hydrogel containing loaded NPs released their content in a controlled and sustained way compared with free drugs or drug-loaded NPs in solution. In vitro cellular studies were performed with a human hepatoma cell line. To evaluate the NPs uptake by tumor cells, NPs were loaded with a fluorescent molecule, coumarin-6, and the NPs uptake by tumor cells was efficiently achieved. In the presence of drug-loaded NLH, a significant decrease in the proliferative activity of tumor cells was observed. The in vivo potential of the drug-loaded NLH was evaluated in a hepatoma tumor-bearing mice model after intratumoral administration. The results showed significant tumor growth inhibition and an extension of the survival time of the tumor-bearing mice compared with free drugs.

Ren et al. (2019) developed a thermosensitive hydrogel composed of PCL and Pluronic 10R5 to treat colorectal peritoneal carcinoma [30]. Poly(lactic acid) (PLA) NPs loaded with oxaliplatin, the clinically used chemotherapeutic agent for this tumor, were incorporated in the hydrogel. To potentiate the oxaliplatin’s activity, tannic acid, a natural compound with anti-cancer properties and lower side effects, was also entrapped in the NPs. The authors verified that while tannic acid showed a similar in vitro release profile from NPs both in solution and in the hydrogel, NPs incorporation slowed the release of oxaliplatin. After intraperitoneal injection in healthy mice, the hydrogel underwent self-gelation at physiological temperature and exhibited a slow degradation rate for 20 days, allowing for a sustained and controlled drug release. Further, a tumor mice model was established by injection of colon cancer cells in the peritoneal cavity of the animals. The incorporation of drug-loaded NPs in the hydrogel improved tumor growth inhibition and increased mice survival compared with the free drugs combination. The NLH also proved to decrease drug toxic effects associated with systemic administration.

Segovia et al. (2015) proposed oligopeptide-terminated poly(β-aminoester) (pBAE) NPs for the small interfering RNA (siRNA) delivery for gene silencing in breast cancer therapy [45]. Since these NPs exhibit rapid degradation limiting their ability to maintain a sustained delivery, these were incorporated in a hydrogel composed of polyamidoamine (PAMAM) cross-linked with dextran aldehyde. In in vitro release studies, the authors verified that NPs’ degradation rate decreased with their immobilization in the hydrogel, increasing stabilization. This experiment also confirmed that NPs incorporation in the hydrogel did not affect the hydrogel properties, such as its degradation rate. The siRNA delivery by NLH enhanced gene silencing ability in human breast cancer cells due to a higher transfection activity than the hydrogel containing free siRNA or siRNA-NPs in solution. In vivo studies were performed in mice bearing tumors in the mammary fat pad implanted with loaded-NLH with disk-shapes (6 mm diameter, 3 mm thick). These studies revealed that NPs exhibited a more sustained and controlled release when intratumorally injected embedded in the hydrogel than in suspension. Although the prepared formulation proved to be safe, the authors did not verify a significant therapeutic effect in the treated animals.

Men et al. (2012) developed NPs composed of pegylated PCL and 1,2-dioleoyl-3-trimethylammonium-propane (DOTAP) for the delivery of deguelin for bladder cancer therapy [46]. Deguelin is a natural compound with anticancer activity but exhibits neurotoxicity. The authors proposed its local delivery to decrease toxic effects by incorporating the loaded NPs in a thermosensitive hydrogel composed of Pluronic F127. In vitro release studies revealed that the hydrogel is gradually degraded, allowing for the sustained release of deguelin. Additionally, in vitro studies also showed that this hydrogel can adhere to the mucous membrane of the bladder wall, thus increasing the drug residence time. Further, to evaluate the ability of the developed NLH formulation in delivering drug cargo to the bladder, mice were intravesically administered with 2 mg/kg hydrogel. The hydrogel was administered in the liquid form, and since its gelation temperature is 25 °C, this formed a gel in situ inside the bladder. The NPs incorporated in the hydrogel were marked with a fluorescent dye (coumarin 6) to allow their visualization. Fluorescent NPs in solution were used as control. The authors verified that although NPs can enhance the drug permeability into the bladder, NPs incorporation in the hydrogel is advantageous since, due to its bioadhesive property, the hydrogel is not eliminated during urination, allowing for a prolonged local drug release without systemic toxicity.

Brachi et al. (2020) proposed polyurethane (PUR) NPs loaded in a Poloxamer 407 thermosensitive hydrogel for delivery to glioblastoma tumors [32]. Local administration can be particularly advantageous to treat brain disorders, such as brain tumors, once NPs can circumvent the blood–brain barrier that poses a significant obstacle for brain drug delivery. The hydrogel formulation was optimized, and it was verified that gelation time decreases with the increase of poloxamer concentration. A polymer concentration of 25 wt.% was chosen to obtain a gelation time of 4 min at physiological temperature. NP incorporation in the hydrogel proved to be advantageous by slowing the cargo release and preventing the burst effect. To assess the ability of the hydrogel to increase the NPs retention time in the tumor, intracranial xenografts were established in immunocompromised mice. BODIPY fluorophore was encapsulated in the NPs to allow for their real-time biodistribution evaluation, and 2.5 mg hydrogels containing the NPs were administered to the animals by intratumoral injection. A group of animals was treated with NPs in solution as a control. NPs incorporation in the hydrogel increased the retention time of NPs at the tumor site, thus avoiding their migration from the tumor and covering larger areas of the tumor compared with NPs in suspension. The authors concluded that these NPs are a promising tool for glioblastoma drug delivery to increase drug accumulation in the tumor without systemic toxicity.

PLGA NPs are biocompatible, Food and Drug Administration (FDA) approved [54], and safe for repeat-dose exposure in vivo [55], being widely explored for incorporation in hydrogels for drug delivery. Wang et al. (2021) developed a hydrogel composed of poly (ethylene glycol) diacrylate (PEGDA) and HA for the local delivery of paclitaxel for lung cancer therapy [26]. The authors encapsulated the drug into PLGA NPs and incorporated them in the hydrogel. Aiming for the in situ ultraviolet (UV) polymerization of the hydrogel at the tumor site, a radical photoinitiator for the UV curing was added to the formulation, and its effect on the hydrogel gelation time, swelling rate, and degradation rate were evaluated. The authors verified that increasing the concentration of the photoinitiation led to a decrease in the hydrogel gelation time, degradation rate, and pore size due to a reduction of the water content. Incorporating the natural polymer HA in the formulation increased the hydrogel’s swelling rate, leading to higher NPs’ loading capacity. In vitro release studies showed that paclitaxel encapsulation in the PLGA NPs allows for a slow and sustained drug release. PLGA NPs have similar release profiles in the hydrogel or in solution. Mice bearing subcutaneous tumor xenografts were treated with paclitaxel-loaded NPs in solution or incorporated in the hydrogel (5 mg/kg). After local injection, the regions of tumors were irradiated with UV light to allow the in situ hydrogel polymerization. The obtained results depicted that incorporating the drug-loaded NPs into the hydrogel improved tumor growth inhibition since the hydrogel can retain the NPs at the tumor site avoiding elimination by the lymphatic system into the systemic circulation.

Shen et al. (2015) also proposed PLGA NPs-loaded thermosensitive hydrogels for the local delivery of paclitaxel for cancer therapy, in this case, to treat pancreatic cancer [47]. In this work, the hydrogel was composed of HPMC, methyl cellulose (MC), sodium alginate (SA), and Pluronic F-127 and F-68. Its degradation speed and gelation time at physiological temperature were optimized to ensure that the hydrogel remained permeable in the tumor tissue with a slow erosion rate. In vitro release studies revealed that incorporating the NPs into the hydrogel significantly delayed paclitaxel release, preventing the burst release. In vitro studies with drug-resistant pancreatic tumor cells suggested that paclitaxel encapsulation in the NLH increases cell uptake due to the ability of the NPs to circumvent the p-glycoprotein pump and the ability of F-127 to increase the cell membrane’ fluidity. Additionally, in vitro studies in a 3D model composed of tumor cells supported in agarose/collagen scaffold were performed. Results showed that drug elimination rates decreased when NPs were incorporated in the hydrogel, leading to higher drug concentrations in the tissue and, consequently, to more efficient inhibition of cell regrowth. Real-time imaging studies in subcutaneous tumor-bearing mice depicted that, after intratumoral injection, NLH remained near the injection site, while control NPs (without hydrogel) were distributed evenly throughout the tumor tissue. NPs incorporated in the hydrogel proved to be more efficient in inhibiting tumor growth than NPs in solution or free paclitaxel, with no toxicity to healthy organs.

Hydrogels containing drug-loaded PLGA NPs were also widely explored for other biomedical applications to enhance therapeutic efficacy by increasing drug-residence time. Kulsirirat et al. (2021) proposed a gelatin-based hydrogel to deliver andrographolide for osteoarthritis therapy [48]. Andrographolide is a natural compound with anti-inflammatory properties but has inadequate absorption, distribution, metabolism, and excretion properties. For the preparation of the NPs, PLGA polymers with different molecular weights and end groups were used. The choice of the adequate formulation was based on the drug encapsulation efficiency. The chosen NPs formulation was incorporated in the hydrogel. In vitro release studies revealed that andrographolide release from the hydrogel is slower when the compound is entrapped in the selected PLGA NPs than the free compound. Healthy mice were treated with different formulations by intra-articular injection, hydrogel containing a free fluorescent dye, dye-loaded NPs, and dye-loaded NLH. Real-time biodistribution of dye was evaluated using a non-invasive in vivo imaging system. The obtained results showed that incorporating NPs into the hydrogel allowed for a long-term sustained release, increasing the retention time in the target tissue maintaining a constant fluorescence intensity in the joint over 8 weeks. The authors concluded that this hydrogel is a suitable approach for the local management of osteoarthritis.

Saygili et al. (2021) developed a hydrogel composed of alginate and PAM for the in situ delivery of the transforming growth factor beta-3 (TGF-β3) for cartilage regeneration [49]. The growth factor was entrapped in PLGA NPs, and the experimental parameters were optimized to yield NPs with suitable dimensions. Three hydrogel formulations were prepared, one containing empty PLGA NPs, the other containing TGF-β3-loaded PLGA NPs, and control hydrogel without NPs. The mechanical strength of the hydrogels was increased when the incorporation of the NPs was performed. The thermal degradation of the hydrogels was not affected by the NPs incorporation. In vitro biodegradability and stability studies showed that the hydrogels retain their mechanical stability under different temperatures and humidity conditions over 3 months, exhibiting a biodegradation rate of 3.5% (*w*/*w*) per week. The three hydrogel formulations proved to be biocompatible in vitro using mice cells. The hydrogels’ performance was further evaluated in a cartilage defect rat model. Cylindrical cartilage defects (1.5 mm in diameter and 1.5 mm in depth) were created in healthy rats, and then the prepared hydrogels were implanted into the defect site. Control animals were left untreated. Animals treated with TGF-β3-NLH showed enhanced tissue repair with newly formed tissue with chondrogenic differentiation and cell proliferation without excessive inflammation. The authors concluded that the developed NLH is suitable for cartilage regeneration due to its ability to mimic the extracellular matrix structure.

Hydrogels have gained popularity for tissue repair and regeneration applications since these can mimic the extracellular matrix, regulate cell processes, and form new tissue. The porous structure of hydrogels promotes cell attachment and proliferation and allows for the diffusion of crucial nutrients [56]. Therefore, other groups have explored PLGA NPs incorporated in hydrogels for tissue regeneration applications. For example, Gong et al. (2020) [50] developed PLGA NPs incorporated in hydrogel for tissue repair after hemorrhagic injury. The proposed hydrogel was composed of keratin and contained bone marrow mesenchymal stem cells. The PLGA NPs incorporated in the hydrogel were loaded with epidermal (EGF) and basic fibroblast (bFGF) growth factors to promote stem cell differentiation. An iron-chelator was also entrapped in the hydrogel to regulate iron levels since iron overload can decrease the success of stem cell therapy. The hydrogel was prepared using keratin with different molecular weights to assemble an outer shell composed of low-molecular-weight keratin and the iron chelator. The stem cells and PLGA NPs were incorporated in the inner core consisting of high-molecular-weight keratin. The outer shell of the hydrogel exhibited a faster in vitro degradation rate than the inner core, which is advantageous by allowing chelation of iron at a faster rate while slowing the release of the growth factors for stem cells differentiation. The authors also verified that the growth factors were released slower when the NPs were loaded in the hydrogel, which allowed avoiding of the burst release effect. In vitro cytotoxicity studies revealed that the hydrogels are biocompatible at a concentration below 100 mg/L, causing no harmful effects to the stem cells. After injecting animals intracranially with the hydrogel, a neurobehavioral evaluation was conducted. The authors verified that the developed formulation improved stem cell differentiation and accelerated neurological recovery.

Other materials have been proposed for tissue regeneration applications, such as chitosan NPs. Ai et al. (2019) proposed collagen hydrogel containing insulin-loaded chitosan NPs to promote the regeneration of the sciatic nerve caused by traumatic injury or some diseases [51]. Although clinically used to regulate blood glucose, recent evidence has shown that insulin possesses neurotrophic activities. Because collagen is abundantly distributed in the sciatic nerve and other peripheral nerves, it was chosen for hydrogel formation. The developed hydrogel exhibited a mean pore size (75–235 µm) suitable for cell attachment and migration. In vitro studies revealed that after incorporating insulin-loaded NPs, the hydrogel’s degradation rate was decreased, allowing for a controlled and sustained release of insulin for 14 days. The prepared formulation did not induce hemolysis in vitro, thus proving the good human blood compatibility of the used materials. Additionally, the insulin-NLH did not show any in vitro cytotoxicity to rat glial cells. Therefore, the authors proceed to animal experiments to evaluate the in vivo regeneration potential of the developed hydrogel. A sciatic nerve injury was created in healthy rats, and 0.5 mL of hydrogel formulations (with or without insulin-loaded NPs) were injected at the lesion site. A group of control animals received no treatment. Histological analysis of untreated animals revealed poor fiber arrangement and damages, such as edema, disintegration of the myelin sheath, degenerated nerve fibers, and fibrosis. In contrast, animals treated with insulin-loaded NLH showed regenerated tissue resembling the normal sciatic nerve. Animals treated with control hydrogel without NPs exhibited regenerated tissue, although to a lower extent than those treated with the hydrogel-containing NPs. Both hydrogel formulations proved to decrease the muscle weight loss in the injured limb and recover motor and sensory functions, with NLH being more efficient. The authors concluded that the synergic effect between collagen hydrogel and insulin-loaded NPs could enhance peripheral nerve regeneration.

Chitosan NPs have also been explored for tissue repair after spinal cord injury. Mahya et al. (2012) developed a hybrid hydrogel composed of alginate and chitosan [52]. A natural compound, berberine, was encapsulated into the chitosan NPs to enhance its ability to promote the growth and reconstruction of damaged neuronal cells. In vitro studies revealed that incorporating the NPs into the hydrogel prolonged the berberine release three-fold, preventing the burst release. The incorporation of the NPs also altered the mechanical properties of the hydrogel by increasing the degree of crosslinking, leading to an increased elastic modulus. The NPs presence also increased the degradation rate of the hydrogel. The degradation of the scaffolds in vivo allows for the formation of new tissue. Stem cells were also incorporated in the scaffold formulation to potentiate the neurodegeneration ability of this hydrogel. In vitro studies revealed that the cells remained viable after their entrapment in the hydrogel. For the in vivo evaluation, healthy rats were submitted to a laminectomy surgery to induce a moderate spinal cord contusion injury. Then, hydrogels were implanted at the lesion sites. The biological performance of different hydrogel formulations was evaluated using a hydrogel with control NPs (without natural compound), a hydrogel with berberine-loaded NPs, and a hydrogel with berberine-loaded NPs and stem cells. A group of animals was left untreated as a control. The authors verified that the berberine-loaded NLH induced successful spinal cord injury recovery contrary to the control hydrogel. Additionally, the hydrogel containing both berberine-NPs and stem cells exhibited the better recovery of sensory and motor functions, suggesting that combination therapy with stem cells is a promising approach for repairing spinal cord injury.

Furthermore, for tissue repair, Gan et al. (2019) have proposed an approach using Ag-Lignin NPs [53]. Ag-Lignin NPs exhibit bactericidal activity due to their ability to generate free radicals, advantageous for wound repair. The NPs were loaded in a hydrogel composed of pectin and polyacrylic acid (PAA) to potentiate their therapeutic effect. The formulations were physicochemically characterized, and the authors verified that Ag-Lignin NPs improved the mechanical properties of the hydrogel and triggered its self-gelation at room temperature, not requiring UV or thermal treatments that could damage tissues. The adhesiveness properties of the hydrogel to different surfaces were evaluated in vitro, and it was verified that the hydrogel maintained good adhesion to porcine skin after 28 days. The NLH exhibited high antibacterial activity in vitro and in vivo, inhibiting the growth of both Gram-negative and Gram-positive bacteria. The biological effect of the hydrogel performance was evaluated in a wound rat model. A skin wound was created in the dorsal area of the animals, and the animals were treated with 30 μg NLH or control hydrogel without NPs. The hydrogels were implanted in the wound site, and control animals were left untreated. Although rats treated with control hydrogel exhibited healed wounds, the NLH showed better healing with higher portions of mature tissue with collagen fibers. Besides its potential for wound healing, this NLH system could also be applied for bone or cartilage repair.

### 4.2. Topical Administration of Polymeric NPs-Loaded Hydrogels

#### 4.2.1. Ocular Administration by Instillation

Due to eye complexity (Figure 4), eye barriers, and their physiological functions, the conventional ocular administration of drugs has several shortcomings in terms of low bioavailability and reduced activity due to a low permanence and permeability at the target site [57].

High dosages and frequent application of drugs are required to achieve the therapeutic drug concentrations, which can induce eye tissue injury, unwanted side effects, and low patient compliance. In the last years, polymeric NLH have been explored as DDS for ocular administration, as summarized in Table 4. These systems can increase the drugs’ bioavailability and retain drugs at the ocular tissues for an extended period of time and release them in a controlled and sustained manner, enhancing the therapeutic effect [58]. Furthermore, with hydrogels, a reduced frequency of administration is required to achieve the therapeutic drug concentration, increasing patients’ compliance.

Different approaches have been developed for glaucoma treatment. For example, Yang et al. (2012) proposed an NLH system to deliver two antiglaucoma drugs, timolol maleate and brimonidine [59]. The authors developed PLGA NPs as drugs carrier integrated into a PAMAM-based hydrogel. This hydrogel presents low viscosity, allowing for its administration in the liquid form as eye instillation. In vitro release studies were performed, showing that the NPs could release both drugs simultaneously for 28 days, while NPs integrated into the hydrogel released them for 35 days. Additionally, studies with human corneal epithelial cells were performed to evaluate the NLH cytotoxicity and the cellular uptake of the NPs. The results proved the non-toxicity of the NLH formulation, posteriorly corroborated with an in vivo analysis. The NPs uptake was about seven times higher when embedded in the hydrogel than in solution. Furthermore, the authors proceeded to animal experiments with ocular normotensive rabbits to evaluate the in vivo therapeutic activity of the developed hydrogel. A topical drop was administrated in the right eye, while the other eye was used as control, and the intraocular pressure (IOP), a parameter that in high values is associated with glaucoma development, was measured for seven days in both eyes. When NPs were embedded in the hydrogel, results showed an extended and controlled release of drugs, reduced IOP, and higher concentrations of drugs in aqueous humor, cornea, and conjunctiva for 4 days. The authors concluded that the NLH could retain drugs at the target tissue, significantly reducing the frequency of administration and improving the patients’ compliance.

Cheng et al. (2019) also proposed PLGA NPs loaded in a thermosensitive chitosan-gelatin-based hydrogel as an eye drop formulation for glaucoma treatment [58]. Recently, oxidative stress in the ocular tissue has been associated with glaucoma. To overcome that, the authors proposed the administration of curcumin, an antioxidant and anti-inflammatory natural compound, encapsulated into NPs. Then, loaded-NPs were incorporated in the hydrogel together with latanoprost. Hydrogel in vitro release study revealed a prolonged release of curcumin-loaded NPs and latanoprost for 7 days. Under oxidative stress, an in vitro cell viability assay was performed with human ocular connective tissue. In the presence of the hydrogel containing loaded NPs, the cell damage caused by the oxidative stress decreased. Additionally, the anti-inflammatory effect of NLH formulation was confirmed by the considerable regulation of the inflammation-related genes compared with control hydrogel. Rabbit corneal epithelial cells were used to evaluate the in vitro biocompatibility of the developed NLH, and the results showed no damaging effect on cells. Thus, in vivo studies were performed with albino rabbits, instilling one eye with the loaded NLH (50 µL) and using the untreated eye as a control. After 7 days, the animals did not present indications of inflammation, confirming the biocompatibility results obtained in vitro.

The same authors developed an approach with the same materials to deliver two drugs with anti-inflammatory properties for inflammatory treatment after intraocular surgery [60]. In this work, the authors proposed levofloxacin encapsulated into PLGA NPs. Thus, NPs were simultaneously encapsulated in the hydrogel with prednisolone acetate. An in vitro release study was performed with drug-loaded NLH formulation, whose results showed a sustained release of drugs for 7 days. The levofloxacin-loaded NPs and prednisolone acetate were both incorporated in the hydrogel, and rabbit corneal epithelial cells were used to evaluate the in vitro cell viability and the anti-inflammatory and anti-bacterial properties of the NLH. Tumor necrosis factor-alpha was used to induce cells’ inflammation. Then, cells were treated in vitro with blank solution, prednisolone acetate loaded-hydrogel, levofloxacin loaded-NPs, or drugs-loaded NLH. The results showed a considerable down-regulation in the expression of inflammation-related genes for all the tested formulations, compared to the blank solution, the levofloxacin-loaded NLH being the most promising. The loaded NLH performance was further evaluated in treating infected corneas of an ex vivo rabbit model of *Staphylococcus*
*aureus* keratitis. The corneas were treated with 50 μL formulation for 24 h, and the results confirmed the anti-inflammation and anti-bacterial properties of the hydrogel. The NLH sustains drugs at the cornea, which increases the ocular bioavailability of drugs, and reduces the frequency of administration. Furthermore, the extended and controlled release of drugs provides a longstanding anti-inflammatory and anti-bacterial treatment, reducing side effects.

Abrego et al. (2015) also used PLGA NPs for the anti-inflammatory treatment after ocular surgery [61]. Pranoprofen is a non-steroidal anti-inflammatory drug commonly used after ocular surgery but with drawbacks in ocular bioavailability, stability, and solubility. PLGA NPs were used to encapsulate pranoprofen to increase its absorptivity in the ocular barrier. Then, two formulations of carbomer 934-based hydrogel with loaded-NPs were prepared for ocular administration: containing 0% and 1% azone as permeation enhancer agent. Both NLH formulations (with and without azone) decreased drug release by about 50% for 24 h compared with free drug or commercial eye drops. In vitro permeation and retention studies through the cornea were performed with ex vivo corneas of healthy rabbits using NLH formulations (with and without azone), free drug solution, and commercial eye drops. The results showed that both formulations increased the drug permeation and retention in the cornea. As expected, the NLH prepared with azone presented better results proving its permeation enhancer properties. The ocular tolerance of the hydrogel was evaluated in vivo through a single instillation of 50 μL of each NLH. Ocular lesions involving animals’ corneas, irises, and conjunctivae were analyzed. Both formulations were considered safe since no indications of opacity, inflammation degree, congestion, swelling, or discharge were observed. In this experiment, the authors also demonstrated the anti-inflammatory efficacy of NLH with azone and its ability to reduce edema, being a suitable approach for ocular administration of pranoprofen.

Ocular administration with polymeric NLH systems can also be advantageous to treat bacterial conjunctivitis. The treatment’s efficacy is hindered by low bioavailability and poor permeability of antibiotics in the ocular tissue. Due to this, higher concentrations of antibiotics are prescribed, which increases bacterial resistance [64]. To overcome this problem, Alruwaili et al. (2020) prepared gentamycin-loaded chitosan NPs integrated into a Carbopol 974P-based hydrogel for bacterial conjunctivitis treatment [62]. The incorporation of NPs in the hydrogel was optimized to obtain suitable properties for ocular delivery and evaluated in terms of clarity, pH, gelling ability, and rheological behavior. Both in vitro and ex vivo studies were performed with NPs in solution, gentamycin-loaded NLH, and commercial eye drops of gentamycin. The NPs-hydrogel system significantly prolonged the release of the drug. The authors verified that both NPs and NLH slowed the drug release compared with the commercial solution, with the NLH exhibiting the slower release. Cornea tissue collected from the goat eye was used for ex vivo studies to evaluate the mucoadhesive strength, ocular permeation, ocular tolerance, and antimicrobial properties of the hydrogel formulation. Outcomes revealed a high mucoadhesive strength indicating a greater retention capacity of the system in the cornea. As the ocular permeation was higher for the NPs in solution, the author suggested that the hydrogel network decreased the permeation. Furthermore, results indicated that the NLH significantly improved the drug’s antimicrobial activity. All the findings indicate that the NLH was suitable for ocular administration of gentamycin by increasing the corneal contact time and extending the drug release. Furthermore, the antimicrobial properties of hydrogel enhance the effectiveness of bacterial conjunctivitis treatment.

Fabiano et al. (2019) proposed a chitosan-based hydrogel and NPs prepared with chitosan derivatives to promote 5-fluorouracil transcorneal administration, a drug widely used for several ophthalmic diseases [63]. NPs were prepared with four different chitosan and its derivatives: chitosan, quaternary ammonium-Ch conjugate (QA-Ch), S-protected derivative thereof (QA-Ch-S-pro), and a sulfobutyl chitosan derivative (SB-Ch). The NPs were optimized to increase the ocular bioavailability of 5-fluorouracil. The hydrogels were prepared with chitosan, loaded-NPs, and β-glycerophosphate disodium salt as a gelling agent. In vitro 5-fluorouracil release was performed with NPs in solution and NLH. All the prepared NPs revealed long drug retention and a sustained release, suitable for corneal delivery. In vitro results of the NLH demonstrated a slower release for hydrogel combined with SB-Ch NPs. SB-Ch is negatively charged and interacts electrostatically with the positive charge of chitosan hydrogel. The rheological analysis further validated this interaction, which revealed a robust mucoadhesive role influenced by NPs charge. In vivo studies were performed with albino rabbits, instilled in one eye with 50 µL of each formulation, using the other untreated eye as a control. NLH were able to increase the bioavailability of the drug and prolong its retention time at the cornea, compared to control. No signs of corneal edema were observed for all the tested formulations. The authors concluded that the formulation with SB-Ch NPs is the most efficient in prolonging the retention of the drug at the cornea, constituting a more effective treatment.

#### 4.2.2. Epidermic Administration

Skin is the largest organ and the first barrier of the human body against external influences [65,66]. Due to this constant exposure to the external environment, skin is constantly subject to wounds, which in turn are quite predisposed to inflammation or infection that make their healing difficult [67]. Beyond that, for patients with other comorbidities, such as diabetes, inflammation could be uncontrollable, and wound healing could turn into a chronic medical problem [68]. Nowadays, the treatments applied to wounds protect the wounds from pathogens and incorporate active pharmaceutical agents to help with the healing process by treating the existent inflammation/infections [69]. Topical administration of drugs is usually used for external treatments and corresponds to the most non-invasive form of drug delivery. However, drug penetration in skin tissue is limited by drugs’ properties, such as high molecular weight or reduced bioavailability [25], which require high dosages of drugs. Furthermore, a frequent application is needed, especially in the case of chronic wound healing, which demands high patient compliance that most of the time is not achieved. These limitations compromise the treatments and could be addressed using polymeric NLH as DDS, as proposed by several authors (Table 5). Besides the NPs improving drugs’ bioavailability, protecting them from degradation, they could also be modified to enhance tissue penetration [70]. Hydrogels offer a favorable environment for wound healing with appropriate humidity, oxygen levels [71], and temperature to the tissue [69]. Hydrogels can also absorb wound exudates and provide impermeability to bacteria, preventing infection [68]. By incorporating NPs loaded with drugs, hydrogels offer a controlled release of drugs after application and increased retention of drugs in the wound area [25], improving the therapeutic effect.

Bernal-Chávez et al. (2020) developed a Pluronic F-127 thermo-responsive hydrogel containing PLGA NPs loaded with a platelet lysate to treat chronic wounds [23]. The hydrogel formulation was optimized, and the gelation temperature was 32 °C, allowing the formulation to solidify in situ and be retained in the wound area during the platelet lysate release period. In vitro release studies revealed that lysate-loaded NPs incorporated into the hydrogel significantly prolonged its release from 12 to 24 h. The therapeutic efficacy of the NLH was evaluated in healthy mice with a full-thickness cutaneous wound in the dorsal region with a diameter of 3 mm. In vivo results demonstrate that the topical application of platelet lysate-loaded NPs incorporated into the hydrogel accelerates wound closure by promoting fibroblasts’ cell migration and proliferation, proving to be more efficient than platelet lysate in solution.

Saleh et al. (2019) proposed a gelatin methacryloyl-based hydrogel containing HA NPs loaded with miR-223 5p mimic (miR-223) to promote wound healing [68]. miR-223 regulates the expression of anti-inflammatory and proinflammatory markers. The developed NLH showed a controlled release of miR-223 over 48 h. In vitro results confirmed that the hydrogel containing miR-223-loaded NPs was successfully internalized by murine macrophages and induced their polarization by decreasing the expression levels of pro-inflammatory markers. The therapeutic efficacy of the system was evaluated in male mice with wounds on the dorsum. According to the visual inspection of the wounds, the percentage of wound closure induced by the hydrogel containing miR-223-loaded NPs was substantially higher than the unloaded hydrogel and the free miR-223. Histological analysis of tissue samples validated these results, revealing that the miR-223-loaded NLH promotes wound healing by initiating the resolution of the inflammatory phase and stimulating the formation of new vascularized skin tissue.

A hydrogel loaded with polymeric NPs was also proposed by Aly et al. (2019) to be topically used on skin wounds [72]. PEG NPs were prepared to encapsulate simvastatin, a molecule that promotes the wound healing process. The hydrogel made of Carbopol showed a controlled release of simvastatin in vitro. Furthermore, an ex vivo permeation study revealed that 69% of the drug permeated through the skin. The therapeutic efficacy of the hydrogel containing loaded NPs was assessed in healthy rats with 8 mm excisional skin wounds on their backs. The wounds treated with the drug-NLH showed a more accentuated reduction in the wound area than those receiving the control hydrogel. In addition, the wounds treated with simvastatin-loaded NLH appeared contracted without hard or abnormal tissue on their surface, unlike control wounds, which showed a hard and abnormal tissue on the surface. The histopathological examination of the medicated wound confirmed the efficacy of the NLH in accelerating wound healing by forming a normal epithelial layer and mature collagen fibers with minimal inflammatory cell infiltration. In contrast, granulation tissue formation with massive inflammatory cell infiltration was observed in the control wound. Hair follicle growth was observed at the end of the treatment indicating good tissue regeneration.

Because open wounds provide an access point for microorganisms, these are particularly vulnerable to bacterial infections. This can result in significant wound inflammation, compromising wound healing and ultimately resulting in death [69]. Thus, keeping a wound from becoming infected is crucial for proper wound healing. Zeng et al. (2021) synthesized a hydrogel composed of xanthan gum and konjac glucomannan to heal bacteria-infected wounds [73]. The hydrogel was loaded with polydopamine NPs. In vitro near-infrared photothermal antibacterial experiments indicate that the NLH has a broad-spectrum antibacterial activity against Gram-negative (*Escherichia coli)* and Gram-positive (*Staphylococcus aureus*) bacteria. While bacteria incubated with the buffer and the unloaded hydrogel had smooth and intact membranes, bacteria cultured with the loaded hydrogel showed many wrinkles and disruptions. The authors further evaluated the in vitro blood compatibility of the hydrogel using mouse-derived fibroblasts. After 3 days of incubation, the hydrogel showed no cytotoxicity to the healthy cells. To further examine the suitability of the NLH for the topical application in skin tissue repair, male rats with a round skin wound of 8 mm diameter on their backs were used. Next, an *Escherichia coli* suspension was added to the wound surfaces to create bacteria-infected wounds. The developed NLH exhibited a rapid wound shape adaptability. The obtained results demonstrate that the NLH significantly accelerates wound healing compared with the control groups. The wound sites of the animals treated with NLH, were histopathologically analyzed. Results revealed greater regularity of both epithelium and connective tissue, improved collagen deposition, promotion of vascular angiogenesis, and new hair follicles in the wound sites of the animals treated with NLH. In addition, the expression of pro-inflammatory cytokines was significantly lower in the animals receiving NLH than the buffer and unloaded hydrogel groups.

Shafique et al. (2021) also devised a hydrogel with antibacterial properties for wound healing. The hydrogel was composed of HA, pullulan, and PVA, containing cefepime loaded into chitosan NPs [66]. Cefepime is a water-insoluble hydrophobic drug that acts as a parenteral antibiotic against Gram-positive and Gram-negative bacteria. The prepared NLH showed good stability and swelling capacity, allowing for the sustained release of cefepime. The antibacterial activity of cefepime-NLH was evaluated against *Staphylococcus aureus*, *Pseudomonas aeruginosa*, and *Escherichia coli*. It was found to inhibit both Gram-positive and Gram-negative bacteria growth. The in vitro cytocompatibility was also assessed against a human fibrosarcoma cell line, showing no cytotoxicity. Authors evaluated the therapeutic efficacy of the cefepime-loaded NLH in healthy rats with a wound on the back of 1 cm^2^ area. The drug-NLH showed a greater potential to accelerate the wound healing process with a wound closure rate of 100% after 14 days, compared to cefepime in solution, with a wound closure rate of 80% at the same period. The data also demonstrated the ability of the blank hydrogel to accelerate wound healing, with a 90% wound closure rate following 14 days of the application. This is related to the HA’s ability to facilitate the migration and proliferation of fibroblasts as well as absorb water, keeping the wound bed wet. Furthermore, the degradation products of HA promote tissue regeneration by meeting the nutritional needs of the surrounding cells. Moreover, pullulan provides energy to the cells, regenerating the skin through glucose consumption. The healing process of the developed NLH was proved by the presence of hair follicles, sweat, and sebaceous glands and the absence of inflammatory cells at the wound site.

A promising approach to treat bacterial infections present either subcutaneously or in open wounds was proposed by Wang et al. (2021) [67]. To this end, a PAM hydrogel containing PAMAM NPs was prepared for the controlled release of platensimycin, a natural antibiotic with great potential to treat infections caused by *Staphylococcus aureus*. The in vitro release data suggest that incorporating the platensimycin-loaded NPs into the hydrogel provided a more controlled release behavior than the free drug and platensimycin-loaded NPs. Furthermore, the in vitro antibacterial activity study revealed that the hydrogel containing platensimycin-loaded NPs could completely inhibit the *Staphylococcus aureus* growth, and to a greater extent than the free platensimycin and the drug-loaded NPs, without inducing toxicity to murine macrophage cells. In the first approach, the authors evaluated the in vivo activity of the drug-loaded NLH against the subcutaneous *Staphylococcus aureus* infection. The buffer, free drug, drug-loaded NPs, blank NLH, and the hydrogel containing the drug-loaded NPs were subcutaneously administered in the backs of male mice, and the number of colonies in each group was quantified. While the free platensimycin and the blank NLH exhibited no antibacterial activity, the drug-loaded NPs, loaded or not in the hydrogel, displayed high anti-staphylococcus activity. Further in vivo data revealed that the NLH can remain in situ for 24 h, unlike the free NPs, suggesting that the hydrogel can improve platensimycin’s therapeutic activity against bacterial infections by extending its residence time. On the second approach, the authors evaluated the drug-loaded NLH ability in promoting the healing of bacteria-infected wounds following topical application. For this purpose, *Staphylococcus aureus* was used to infect wounds of 10 mm diameter formed on the back of healthy rats. The platensimycin-loaded NLH showed a significantly higher wound healing rate on the ninth day when compared to the other groups. The NLH’s antibacterial activity was assessed after two days by quantifying the remaining colonies in the wound. The hydrogel containing drug-loaded NPs exerted the strongest antibacterial activity in infected wounds than the free drug and the drug-loaded NPs, implying that the NLH is more advantageous for treating bacteria-infected wounds due to the sustained drug release.

In addition to open wounds, bacteria can enter the body through other routes, including mouth, eyes, nose, or urogenital openings. Frequently, bacterial infections occur at sites with high shear forces, which facilitate bacterial adhesion and prevent effective drug accumulation [76]. Thus, developing formulations that can withstand strong shear forces and work at infection sites involving the shear flow of biological fluids is particularly desirable. To address this issue, Zhang et al. (2016) designed a bioadhesive NLH for the local delivery of ciprofloxacin, a broad-spectrum antibiotic [74]. Ciprofloxacin-loaded PLGA NPs were embedded in a hydrogel composed of acrylamide, PEG dimethacrylate, and PVA. Dopamine methacrylamide, a catechol moiety responsible for the adhesion of marine mussels to various surfaces, was included in the hydrogel network. The authors started to compare the in vitro ciprofloxacin-loaded NLH release profile to that of ciprofloxacin-loaded hydrogel (without NPs). While the drug showed a burst release profile from the blank hydrogel, the NLH showed a gradual drug release profile, highlighting the benefit of introducing NPs into the hydrogel to allow the controlled and sustained ciprofloxacin release. Under the flow environment, the in vitro antibacterial efficacy of the formulations (free drug, drug-NPs, blank hydrogel, and drug-loaded NLH) was investigated. The ciprofloxacin-NLH completely inhibited the formation of an *Escherichia coli* bacterial film, unlike the other formulations. The authors then tested the adhesive capabilities of the hydrogel and its ability to hold the embedded NPs under flow conditions, using distinct surfaces including a bacterial film, a mammalian cell monolayer, and a shaved mouse skin tissue. The retention of NPs in the bioadhesive hydrogel was quantified and revealed that the totality of the NPs remained on the three biological surfaces. In contrast, a small quantity of NPs in the non-adhesive hydrogel was retained. Lastly, the NLH’s skin toxicity was investigated in healthy mice following its daily topical application for 7 days, and the results showed no skin reaction or toxicity.

Aiming to improve current psoriasis’ therapy, Mao et al. (2017) [75] designed a silk fibroin hydrogel containing RRR-α-tocopheryl succinate-grafted-ε-polylysine NPs for the topical delivery of curcumin for psoriasis’ treatment. The in vitro release study revealed that the encapsulation of curcumin into the polymeric NPs induced a slower release profile with no evident initial burst release, unlike the free curcumin. The incorporation of the NPs in the hydrogel resulted in a longer-lasting release of the natural compound. The in vivo skin penetration of the curcumin-loaded hydrogel, with and without NPs, was investigated by applying the formulations topically on the back of mice skin (1 cm^2^ area). A high curcumin skin permeation ability was observed when the NPs were included in the hydrogel. The in vivo therapeutic effect of the two formulations was evaluated on a psoriatic mice model and compared to the clobetasol’s activity, a topical corticosteroid used on the psoriasis therapy. Clobetasol-treated mice exhibited the most improvement in psoriatic symptoms, including erythema, thickness, and scaling of the back skins. Although to a lesser extent, curcumin-loaded NPs also improved the psoriatic symptoms, which were further enhanced by incorporating the NPs into the hydrogel. The histological examination confirmed the visual observation as a decrease in thicknesses of the dermis and epidermis—associated with psoriasis—was detected. Significantly few leukocytic infiltrations were noticed in the curcumin-loaded NPs treated group. This suppression was even more evident in the curcumin-loaded NLH group. Finally, the effect of the curcumin-loaded hydrogel, with and without NPs, on inflammatory cytokine levels was evaluated and compared to clobetasol, a corticosteroid used to treat psoriasis. Curcumin-loaded NLH inhibited the expression of inflammatory cytokines to the same degree as the commercially available clobetasol but to a greater extent than the curcumin-loaded hydrogel without NPs.

The topical application of hydrogels containing polymeric NPs has been used to manage other health conditions, such as autoimmune disorders. Rheumatoid arthritis therapy was recently addressed by Khan et al. (2021) [65] by developing a pH-responsive NLH to deliver ibuprofen, a nonsteroidal anti-inflammatory drug widely employed to treat this condition. Eudragit L 100 polymer was chosen to produce the NPs due to its pH-responsive dissolution profile, as it particularly releases drugs at pH 6.8, a pH found in inflamed tissues. The NPs were incorporated into a Carbopol 934 hydrogel containing argan oil as a permeation enhancer agent. The in vitro ibuprofen release data at pH 6.8 indicate that the NPs caused a more sustained release pattern than the free drug, which was further sustained by incorporating the NPs into the hydrogel. On the other hand, both formulations induced a residual drug release at the physiological pH of the skin (5.5). The ex vivo mice skin permeability study revealed that the hydrogel containing ibuprofen-loaded NPs showed a higher permeation of the NPs than a commercially available ibuprofen cream. The presence of argan oil in the formulation further increased the skin permeability of the NPs 14-fold. The in vivo safety of the ibuprofen-loaded NLH was assessed in healthy mice. The formulation caused no visual signs of skin toxicity, with the histological analysis showing no harm to skin tissues. Further, behavioral experiments and biochemical analysis demonstrated the in vivo therapeutic efficacy of the hydrogel in both acute and chronic inflammatory pain mice models, which was significantly improved compared to the group treated with the marketed cream. The prepared formulation was also able to inhibit the inflammatory processes and oxidative stress. In addition, both bone erosion and soft-tissue edema in the ankle joints of the treated mice were considerably reduced in the group treated with the hydrogel containing ibuprofen-loaded NPs.

#### 4.2.3. Vaginal Administration

Local vaginal therapy is a non-invasive drug administration route used to create a local pharmacological impact while avoiding systemic exposure. However, the presence of degradative enzymes and vaginal fluid, which reduces medication bioavailability and residence time, decreases the efficacy of vaginal administration. The vaginal fluid also induces a leak of drugs, requiring repeated applications, that may result in low patient compliance [36]. Polymeric NLH systems for vaginal drug administration are promising to overcome the highlighted drawbacks. The mucoadhesive ability of hydrogels provides a high interaction with vaginal tissue, favoring formulation permanence on the vaginal mucosa. This capability lengthens drug residence time, maximizing pharmacological activity. The combination of NPs and hydrogels provides a high concentration of drugs at the site of action and controlled drug release rates, which decreases the need for frequent applications [77].

Recently, Zimmermann et al. (2021) [78] produced an NLH system to treat vulvovaginal candidiasis. The approach containing gellan gum hydrogel and PCL NPs was used for diphenyl diselenide delivery. Diphenyl diselenide exhibit a wide range of biological effects, including antioxidant, anti-inflammatory, and antifungal activity against *Candida* spp. In vitro assays performed against various *Candida* species confirmed the diphenyl diselenide’s antifungal activity and revealed that the encapsulation of the drug into the NPs did not affect its therapeutic activity. The in vivo efficacy of the formulations (hydrogel containing free or encapsulated drug) was assessed in a mice model of vulvovaginal candidiasis. The formulations were topically administered once a day for 7 days, and the total fungal burden was quantified. Treatment with both formulations reduced the fungal load, validating the diphenyl diselenide’s antifungal activity previously demonstrated in vitro. Additionally, the hydrogel containing drug-loaded NPs presented better pharmacological efficacy than the hydrogel containing free diphenyl diselenide, and the control group treated with Nystatin cream, a commercially available antifungal medicine.

## 5. Discussion and Conclusions

Until this date, the use of polymeric NLH has been studied for different routes of administration, including parenteral and topical administration. As demonstrated in the reported works, the described systems validate the usefulness of polymeric NLH as DDS for various biomedical applications. Based on the in vivo findings, polymeric NLH systems have shown the ability to regulate drug release kinetics and increase the drug’s bioavailability, prolonging the therapeutic effect while minimizing systemic spread and toxicity. The innovative integration of NPs in hydrogels has resulted in a new second generation of DDS with improved performance and characteristics. The target and localized drug delivery offered by NPs could be enhanced through simultaneous use of hydrogels, demonstrating superior capabilities on local drug retention. The presented studies have highlighted the crucial role of polymeric NLH in the reduction of drugs administration frequency, resulting in enhanced patient compliance. Polymeric NLH appear to be a promising strategy to improve the treatment of various diseases by boosting drug delivery efficiency. Two different approaches of polymeric NLH could be employed, NPs loaded into unloaded hydrogels or NPs loaded into the hydrogel, which contains an additional drug. That way, a synergic effect of different drugs could also be obtained.

Due to the extensive availability of materials to produce NPs and hydrogels, many polymeric NLH systems can be generated for a broad range of potential biomedical applications. PLGA is undoubtedly the most explored polymeric material to produce NPs to be incorporated in hydrogels, accounting for about 33% of the works reported above. PLGA is a versatile polymer that can be customized in terms of biodegradation and release rates, which is ideal for drug delivery. PLGA is also biocompatible, biodegradable, and FDA-approved for biomedical use [8]. Chitosan NPs have also been widely studied and were reported in 15% of the works described in this review. Additionally, chitosan is also the most popular material for hydrogel production (18% of the works reported here). This popularity is given by the physicochemical and biological properties of chitosan, which is useful for biomedical applications. In addition to being a natural polymer, it is also non-toxic, biodegradable, and biocompatible. As described previously by Fabiano et al. (2019) [63], chitosan has a cationic characteristic that allows it to interact with the negatively charged cell membranes of microorganisms, conferring to this polymer powerful antimicrobial and antibacterial effects [79]. Other materials have also been explored for hydrogels production, with a particular focus on surfactants, such as Pluronic F-127 and F-68 (12%). Pluronic block copolymers are commonly used in the manufacturing of hydrogels due to their thermosensitive properties. Furthermore, these copolymers can improve the pharmacokinetics and pharmacodynamics of DDS [80].

Although many DDS have considerable in vitro and in vivo drug delivery benefits, scaling up production is still a challenge. Despite that, clinically available strategies of NPs and hydrogels as individual DDS could already be found on the market. The polymeric NPs that have resulted in commercial products are Abraxane, a paclitaxel delivery in a suspension of albumin particles, for breast cancer treatment [81]; Lupron Depot, Nutropin Depot, and Vivitrol, injectable medicines for extended drugs release; and Accurins, for targeted cancer therapy [8]. There is an exhaustive list of commercial products based on hydrogels for biomedical applications. Some examples of commercial products are Metrogel Vagina, a vaginal drug delivery form of the synthetic antibacterial agent, metronidazole; Voltaren Gel, a topical application form of diclofenac used to reduce pain and inflammation; Astero, a topical application form indicated for wounds, first and second-degree burns, post-surgical incisions, cuts, and abrasions; Ocusert, an ocular administration system used for glaucoma treatment; among others [82].

However, their combination as a single DDS is recent, and no clinical trials are concluded nowadays. Just one experiment reporting the use of silk particles distributed in a hydrogel as a tissue filler is in an ongoing clinical trial (NCT04534660). That combination of particles with a hydrogel, called SMI-01, is an injectable formulation applied in the deep dermis, and it is used to correct moderate to severe wrinkles and folds. Furthermore, the researchers are testing the effect of that technology to correct age-related volume deficiency in the midface through either a subcutaneous, supraperiosteal, or both, injection of SMI-01. This feasibility study is multicentered (two clinical sites), unblinded, with no control group, performed to access the preliminary safety and effectiveness of SMI-01 as a tissue filler. The administration of NLH takes place on day 1 and, if desired, on day 30, with the primary safety and effectiveness evaluation taking place at the second month. Extended follow-up examinations will be conducted on subjects for 24 months after the first administration. This clinical trial is still in recruitment status, so there are no reported data on its effectiveness.

Thus, even with the promising advantages of this innovative type of DDS, more research is needed before it reaches the market.

## Figures and Tables

**Figure 1 polymers-14-01010-f001:**
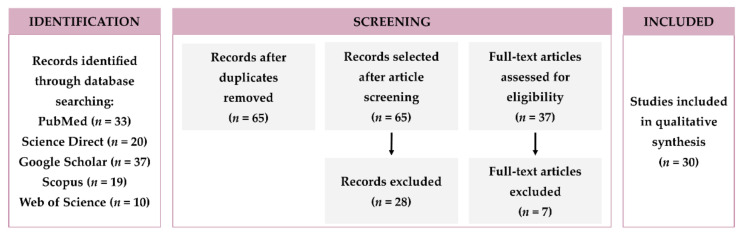
Schematic representation of the applied methodology.

**Figure 2 polymers-14-01010-f002:**
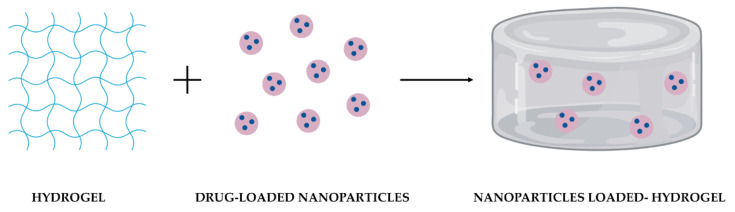
Schematic representation of the combination of drug-loaded NPs and a hydrogel as DDS.

**Figure 3 polymers-14-01010-f003:**
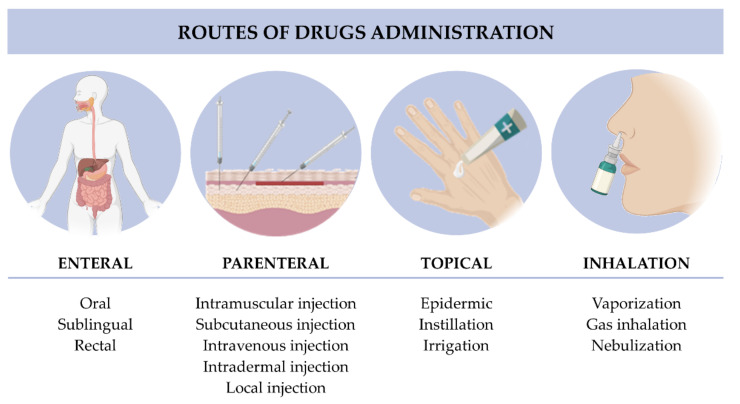
Classification of routes of drugs administration.

**Figure 4 polymers-14-01010-f004:**
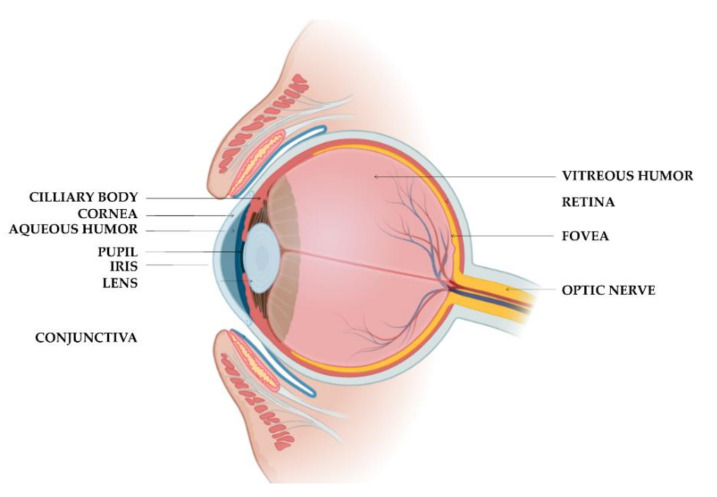
Schematic representation of the human eye’s anatomy.

**Table 1 polymers-14-01010-t001:** Benefits of polymeric NPs, hydrogels, and polymeric NLH as DDS. (+) and (–) indicate advantages and disadvantages, respectively.

	Polymeric NPs	Hydrogel	Polymeric NLH	Refs.
Multiple drug loading	+	+	Improved	[17,20,28]
Hydrophobic drugs loading	+	−	Maintained	[13,17,20,31]
Controlled and sustained release	+	+/−	Improved	[13,28]
Drug bioavailability improvement	+	+	Improved	[13]
Targeted drug delivery	+	−	Maintained	[20]
Local retention of drug	−	+	Maintained	[1,15,30,32]
Stimuli-responsive behavior	+	+	Improved	[12,25,29]

**Table 2 polymers-14-01010-t002:** Summary of polymeric NLH administered subcutaneously for biomedical applications.

NPs Material	Hydrogel Material	Loaded Cargo	NPs Production Method	Hydrogel Crosslinking Nature	Biomedical Application	Main Conclusions	Ref.
PHBHHx	Chitosan	Insulin	Single-emulsion solvent-evaporation	Physical (β-GP)	Diabetes	NLH increased insulin bioavailability and prolonged hypoglycemic effect	[40]
PLGA	PCL–PEG–PCL	ICG and l-Arginine	Double-emulsion solvent-evaporation	Chemical	Various types of cancer	NLH increased ICG and l-Arg concentration and retention at the tumor site, inhibiting tumor growth and regression of the established tumors	[28]
Micelles of PEG-phenylboronic acid-polycarbonate	P(Bor)5-PEG-P(Bor)5 and P(Gu)5-PEG-P(Gu)5	BTZ	Film hydration	Chemical	Cancer (myeloma)	BTZ-loaded NLH enhanced anti-cancer activity by decreasing the tumor size and inhibiting its progression	[41]

**Table 3 polymers-14-01010-t003:** Summary of polymeric NLH locally administered for biomedical applications.

NPs Material	Hydrogel Material	Loaded Cargo	NPs Production Method	Hydrogel Crosslinking Nature	Biomedical Application	Main Conclusions	Ref.
Ethyl cellulose	Chitosan	Carboplatin	Double-emulsion solvent-evaporation	Physical (dibasic sodium phosphate)	Various types of cancer	NLH reduced systemic toxicity, increased drug concentration at the tumor site, and improved anti-tumor activity	[43]
PCL-PEG-PCL	Pluronic F-127	Norcantharidin	Thin-film dispersion	Chemical	Cancer (hepatocellular carcinoma)	NLH provided high anti-tumor activity, with inhibition of the implanted tumors growth and prolonged the survival time of the tumor-bearing mice	[44]
PLA	PCL and Pluronic 10R5	Oxaliplatin and Tannic acid	Double-emulsion solvent-evaporation	Chemical (Sn(Oct)2)	Cancer (colorectal peritoneal carcinoma)	NPs incorporation in the hydrogel allowed for a sustained release in vivo, improving tumor growth inhibition while reducing systemic toxicity	[30]
pBAE	PAMAM crosslinked with dextran aldehyde	siRNA	Self-assembly	Chemical	Breast cancer	The formulation exhibited a sustained and controlled release in vivo, but the therapeutic effect was not improved	[45]
PCL-PEG and DOTAP	Pluronic F127	Deguelin	Film hydration	Chemical	Bladder cancer	NLH acts as a drug depot, allowing for sustained local drug delivery	[46]
PUR	Poloxamer 407	BODIPY (mimic)	Nanoprecipitation	Physical (saline solution)	Glioblastoma	NPs incorporation in the hydrogel increased drug retention time in the tumor tissue, without systemic toxicity	[32]
PLGA	PEGDA and HA	Paclitaxel	Single-emulsion solvent-evaporation	Physical (UV light)	Lung cancer	Incorporating the drug-loaded NPs into the hydrogel improved in vivo tumor growth inhibition	[26]
PLGA-PEG	Pluronic F-127, Pluronic F-68, HPMC, MC and SA	Paclitaxel	Single-emulsion solvent-evaporation	Chemical	Pancreatic cancer	NPs incorporation in the hydrogel increased drug retention time in the tumor tissue, improving tumor growth inhibition.	[47]
PLGA	Gelatin	Andrographolide	Single-emulsion solvent-evaporation	n.d.	Osteoarthritis	NPs incorporation in the hydrogel increases the retention time in the joint, maintaining a sustained release for over 8 weeks	[48]
PLGA	Alginate and PAM	TGF-β3	Nanoprecipitation	Chemical (PAAm crosslinker)	Tissue Regeneration (cartilage)	TGF-β3-loaded NLH induces the formation of new cartilage tissue	[49]
PLGA	Keratin	EGF and bFGF	Double-emulsion solvent-evaporation	Physical (hydrogen peroxide)	Intracerebral hemorrhage (iron overload)	The NLH improved stem cell differentiation and accelerated neurological recovery in vivo	[50]
Chitosan	Collagen	Insulin	Ionic gelation	Chemical (EDC)	Tissue regeneration (peripheral nerve)	Collagen hydrogel has tissue regeneration ability, but the incorporation of insulin-loaded NPs enhances the effect	[51]
Chitosan	Alginate and Chitosan	Berberine	Ionic gelation	Physical (β-GP)	Spinal cord injury	The hydrogel containing berberine-loaded NPs and stem cells exhibited a higher tissue regeneration ability	[52]
Ag-Lignin	Pectin and PAA	None	Self-assembly	Chemical	Wound healing	NPs incorporation in the hydrogel improves wound healing ability-enhancing the formation of mature tissue	[53]

n.d.: not defined.

**Table 4 polymers-14-01010-t004:** Summary of polymeric NLH ocular administered for biomedical applications.

NPs Material	Hydrogel Material	Loaded Cargo	NPs Production Method	Hydrogel Crosslinking Nature	Biomedical Application	Main Conclusions	Ref.
PLGA	PAMAM	Brimonidine and Timolol maleate	Single-emulsion solvent-evaporation	Chemical	Glaucoma	NLH provided a controlled release of drugs, reduction of IOP, and higher concentrations of drugs at the target site.	[59]
PLGA	Chitosan and Gelatin	Curcumin and Latanoprost	Single-emulsion solvent-evaporation	Physical (β-GP)	Glaucoma	The loaded-NLH reduced the oxidative stress effect that causes glaucoma, provided an anti-inflammatory effect.	[58]
PLGA	Chitosan and Gelatin	Levofloxacin and Prednisolone acetate	Single-emulsion solvent-evaporation	Physical (β-GP)	Anti-inflammatory treatment following surgery	Incorporating NPs into a hydrogel, a longstanding anti-inflammatory and anti-bacterial treatment were obtained, reducing side effects.	[60]
PLGA	Carbomer 934	Pranoprofen	Solvent displacement	Chemical	Anti-inflammatory treatment following surgery	NLH provided therapy with improved anti-inflammatory effects and edema reduction.	[61]
Chitosan	Carbopol 974P	Gentamycin	Ionotropic gelation	Chemical	Ophthalmic bacterial infections	NLH increased drug contact time in the cornea, extended release, and excellent antimicrobial properties.	[62]
Chitosan	Chitosan or its derivatives (TSOH)	5-fluorouracil	Self-assembly	Physical (β-GP)	Several ophthalmic diseases	NLH increased drug bioavailability and prolonged drug retention at the cornea.	[63]

**Table 5 polymers-14-01010-t005:** Summary of polymeric NLH epidermally administered for biomedical applications.

NPs Material	Hydrogel Material	Loaded Cargo	NPs Production Method	Hydrogel Crosslinking Nature	Biomedical Applications	Main Conclusions	Ref.
PLGA	Pluronic F-127	Platelet lysate	Double-emulsion solvent-evaporation	Chemical	Wound healing	NLH accelerates wound closure by promoting the cell migration and proliferation of fibroblasts	[23]
HA	Gelatin and methacryloyl (GelMA)	miR-223 5p mimic	n.d.	Chemical	Wound healing	Promotion of wound healing by initiating the resolution of the inflammatory phase and stimulating the formation of new vascularized skin tissue	[68]
PEG 4000	Carbopol	Simvastatin	Nanoprecipitation	Chemical	Wound healing	Acceleration of the wound healing by forming a normal epithelial layer and mature collagen fibers, with minimal inflammatory cell infiltration	[72]
Polydopamine	Xanthan gum and Konjac glucomannan	-	Nanoprecipitation	Chemical	Wound healing	NLH significantly accelerates the healing of wounds by reducing the inflammatory response and promoting vascular reconstruction	[73]
Chitosan	HA, pullulan and PVA	Cefepime	Ionic gelation	Physical (sodium tripolyphosphate)	Wound healing	Accelerates the wound healing process by inhibiting Gram-positive and Gram-negative bacteria growth, with no cytotoxicity against a human cell line	[66]
PAMAM	PAM	Platensimycin	Double-emulsion solvent-evaporation	Physical (PEG di-methacrylate)	Wound healing and subcutaneous bacterial infections	Accelerates wound closure and treats subcutaneous infections by exhibiting antibacterial activity	[67]
PLGA	Acrylamide, PEG dimethacrylate and PVA	Ciprofloxacin	Double-emulsion solvent-evaporation	Physical (PEG di-methacrylate)	Bacterial infections	The bioadhesive NLH showed superior adhesion and antibiotic retention under high shear stress, with no skin toxicity	[74]
RRR-α-tocopheryl succinate-grafted-ε-polylysine	Silk fibroin	Curcumin	Self-assembly	Chemical	Psoriasis	NPs incorporation in the hydrogel improved the therapeutic effect of curcumin by inhibiting skin inflammation	[75]
Eudragit L 100	Carbopol 934 and argan oil	Ibuprofen	Nanoprecipitation	Physical (glutaraldehyde)	Rheumatoid arthritis	Incorporating the drug-loaded NPs into the hydrogels improved the anti-inflammatory effect of ibuprofen compared to the commercially available ibuprofen cream	[65]

n.d.: not defined.

## Data Availability

Not applicable.

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
