# Peer review of "Polymeric Nanoparticles-Loaded Hydrogels for Biomedical Applications: A Systematic Review on In Vivo Findings"

_polymers, 2022, doi:10.3390/polym14051010_

Round 1
Reviewer 1 Report
This manuscript reviewed polymeric nanoparticles-loaded hydrogels for biomedical applications based on routes of drugs administration. The review paper is significant and can be published after a revision. My suggestions are below.
- It is better that authors revise the title of manuscript, changing a systematic review into a short review, as only 60 references appeared in this review.
- Please add the references for Table 1.
- The discussion and conclusion are not in-depth and insufficient.
Author Response
This manuscript reviewed polymeric nanoparticles-loaded hydrogels for biomedical applications based on routes of drugs administration. The review paper is significant and can be published after a revision. My suggestions are below.
Author response: Thank you for your comment. The reviewer’s suggestions were addressed in the revised version of the manuscript. The lines referred in the answers can be found highlighted in the "track changes" version of the manuscript (pdf file).
It is better that authors revise the title of manuscript, changing a systematic review into a short review, as only 60 references appeared in this review.
Author response: Thank you for your suggestion. We have considered a systematic review because we have performed the key stages to produce a systematic review, as according to Pollack et al (2018), such as: clarification of aims and methods, find of relevant research, data collect, study quality assess, synthesis of evidence, and interpretation of findings. We have followed the PRISMA guide (Welch et al. 2016). Thus, we have added a new chapter to the review, chapter 2 (page 2), which explains the methodology used to develop this systematic review.
Refs:
Pollock A, Berge E. How to do a systematic review. Int J Stroke. 2018;13(2):138-156. doi:10.1177/1747493017743796
Welch, V., et al., Extending the PRISMA statement to equity-focused systematic reviews (PRISMA-E 2012): explanation and elaboration. Journal of Clinical Epidemiology, 2016. 70: p. 68-89.
Please add the references for Table 1.
Author response: Suggestion accepted. Please see the new Table 1 (page 5).
The discussion and conclusion are not in-depth and insufficient.
Author response: Suggestion accepted. Please see the new “discussion and conclusion” section (page 24).

Reviewer 2 Report
The presentation of all abbreviations used (so called nomenclature) must be at the beginning of the article.
As the review refers to „Polymeric nanoparticles-loaded hydrogels for biomedical applications” it would have been useful to include the methods of loading the nanoparticles with the bioactive compounds at least in the tables presented, and as well the crosslinking nature for the hydrogels preparation for the examples presented.
Also, the re-verification by spelling and grammar evaluation of the article is also required, as for example: Other materials, such as Ag-Lignin NPs have been proposed for tissue repair, by Gan 517 et al. (2019) [41].
The article reviews some structures based on polymeric nanoparticles-loaded hydrogels for biomedical applications emphasizing more the efficiency in the use of the presented compounds and less on the relationship between the polymeric nanoparticles and the hydrogel network host for the nanoparticles.
Author Response
The presentation of all abbreviations used (so called nomenclature) must be at the beginning of the article.
Author response: Suggestion accepted. Please see all abbreviations in page 1.
As the review refers to „Polymeric nanoparticles-loaded hydrogels for biomedical applications” it would have been useful to include the methods of loading the nanoparticles with the bioactive compounds at least in the tables presented, and as well the crosslinking nature for the hydrogels preparation for the examples presented.
Author response: Suggestion accepted. The tables 2 (page 7), 3 (page 8), 4 (page 16) and 5 (page 19) were revised and new columns with the methods of loading the nanoparticles with the bioactive compounds, as well the crosslinking nature for the hydrogel’s preparation were added.
Also, the re-verification by spelling and grammar evaluation of the article is also required, as for example: Other materials, such as Ag-Lignin NPs have been proposed for tissue repair, by Gan 517 et al. (2019) [41].
Author response: Suggestion accepted. The manuscript was revised accordingly.
The article reviews some structures based on polymeric nanoparticles-loaded hydrogels for biomedical applications emphasizing more the efficiency in the use of the presented compounds and less on the relationship between the polymeric nanoparticles and the hydrogel network host for the nanoparticles.
Author response: Thank you for your suggestion. We used the discussion to emphasize the role of NPs loaded in hydrogels as DDS. Please see the improved discussion and conclusion section in page 24.

Reviewer 3 Report
The Introduction should be improved by describing the advances in nanotechnology applications, including regulamentations and related references should be mentioned such as:
Souto et al. Nanopharmaceutics: Part I-Clinical Trials Legislation and Good Manufacturing Practices (GMP) of Nanotherapeutics in the EU. Pharmaceutics. 2020, 12(2), pii: E146. doi: 10.3390/pharmaceutics12020146
Introductory lines to describe the main sections of paper should be added.
The novelty character of this review respect to others present in literature should be marked.
A section methodology to indicate the criteria of bibliographic research should be added.
Author Response
The Introduction should be improved by describing the advances in nanotechnology applications, including regulamentations and related references should be mentioned such as:
Souto et al. Nanopharmaceutics: Part I-Clinical Trials Legislation and Good Manufacturing Practices (GMP) of Nanotherapeutics in the EU. Pharmaceutics. 2020, 12(2), pii: E146. doi: 10.3390/pharmaceutics12020146
Author response: Thank you for your suggestion. The introduction was improved, and related references were added. Please see lines 36-59, page 2, and the new references [2-8].
Introductory lines to describe the main sections of paper should be added.
Author response: Suggestion accepted. Please see lines 74-80 in page 2.
The novelty character of this review respect to others present in literature should be marked.
Author response: Suggestion accepted. Please see lines 69-74 in page 2.
A section methodology to indicate the criteria of bibliographic research should be added.
Author response: Thank you for your suggestion. The new section “Research Methodology” which explains the methodology used to develop this systematic review was added (page 2).
